# Hope and Technology: Other-Oriented Hope Related to Eye Gaze Technology for Children with Severe Disabilities

**DOI:** 10.3390/ijerph16101667

**Published:** 2019-05-14

**Authors:** Patrik Rytterström, Maria Borgestig, Helena Hemmingsson

**Affiliations:** 1Division of Nursing Science, Department of Social and Welfare Studies, Faculty of Health Sciences, Linköping University, 601 74 Norrköping, Sweden; 2Faculty of Medicine and Health, School of Health Sciences, Örebro University, 702 81 Örebro, Sweden; Maria.Borgestig@oru.se; 3Department of Special Education, Stockholm University, 106 91 Stockholm, Sweden; helena.hemmingsson@specped.su.se; 4Division of Occupational Department of Social and Welfare Studies, Faculty of Health Sciences, Linköping University, 601 74 Norrköping, Sweden

**Keywords:** eye gaze control technology, technology, disabled children, self-help devices, phenomenological-hermeneutic

## Abstract

Introducing advanced assistive technology such as eye gaze controlled computers can improve a person’s quality of life and awaken hope for a child’s future inclusion and opportunities in society. This article explores the meanings of parents’ and teachers’ other-oriented hope related to eye gaze technology for children with severe disabilities. A secondary analysis of six parents’ and five teachers’ interview transcripts was conducted in accordance with a phenomenological-hermeneutic research method. The eye gaze controlled computer creates new imaginations of a brighter future for the child, but also becomes a source for motivation and action in the present. The other-oriented hope occurs not just in the future; it is already there in the present and opens up new alternatives and possibilities to overcome the difficulties the child is encountering today. Both the present situation and the hope for the future influence each other, and both affect the motivation for using the technology. This emphasises the importance of clinicians giving people opportunities to express how they see the future and how technology could realise this hope.

## 1. Introduction

This study seeks to gain a deeper understanding of the hope that parents and teachers might experience when new assistive technology is introduced for children with severe disabilities. Hope is a central concept in health care, but lacks conceptual clarity since there are different types and levels of hope [1] and hope is highly contextual [2]. It is often described from an individualistic perspective (or, self-oriented hope), but it is also possible to have other people as the target for one´s hope, which Howell and Buro [3] call ‘other-oriented hope’. Empirical research shows how other-oriented hope may have a beneficial outcome for both the person who has hope and the targets of that hope, and it may also have positive consequences for their relationships. Parents of children with autism showed a positive correlation between the other-oriented hope they had for their children, and their personal parental hope [4]. Moreover, they also experienced greater life satisfaction, less parenting stress and less depression [5]. A study of parental hope involving children with cystic fibrosis showed that the degree of other-oriented hope is related to the levels of emotional distress, depression and anxiety experienced [6]. Hope is essential for parents when their children are in vulnerable situations. In the process of keeping hope alive, the parent’s fear of losing hope would be like giving up on their child [7]. Hopeful thinking helps parents to cope in their process of making critical medical care decisions for their children with complex conditions [8]. 

Other-related hope is often understood from definitions of self-oriented hope and related theoretical frameworks and based on these characteristics of general hope. Howell [9] defines other-oriented hope as “future-oriented belief, desire, and a mental imagining surrounding a valued outcome of another person that is uncertain but possible” (p. 20). Olsman, et al. [10] describe other-oriented hope as including at least three perspectives: the realistic perspective (which means adjusting hope to be realistic), the functional perspective (which means fostering hope), and the narrative perspective (which means focusing on interpreting hope). This last narrative perspective is this study´s theoretical framework, and in line with hope as an imaginary act, we imagine that thing we are hoping for. Imagination is not a feeling, more a state of mind that imbues one’s entire mood and all one’s emotions. It is a way out of the present situation, imagining a future not yet seen [11]. In this narrative perspective, hope is related to how the human being tries to create a meaningful and worthwhile life in the midst of suffering [12]. Hope is thus as much future as present, a motivation to action, anticipating an active uniting with an uncertain future [13].

Society places a great deal of hope and trust in technology and its impact on the future, not least in the health care sector in which new health technology creates an expectation and a hope of alleviating suffering, curing diseases and creating new opportunities for the future. At the same time, rapid technological development in combination with commercial motivations to get products to market can lead to technology being introduced in the health care sector without evidence that the promised benefits can be realised. The technology may fall short of fulfilling its clinical promises or may be a waste of health care resources [14]. In addition to technology in itself creating hope, research emphasises that the successful implementation of assistive technology to fulfil hope is dependent on how the technology works in everyday life, its interplay with the surrounding environment [15], individuals’ attitudes to technology [16], and clinicians’ training, support and attitudes [17]. A successful implementation of assistive technology has to take contextual factors into consideration, and therefore more knowledge is needed about how people experience and are affected by new technology [18].

One example of promising advanced assistive technology is eye gaze controlled computers. For people with significant physical disabilities and with little or no functional speech, reliable eye movements can provide the possibility to control a computer with their eyes. An infrared sensor tracks the movement of the eyes and translates this into cursor control. In this way, the user can control the position of the cursor on the computer screen through eye movements, without the aid of a mouse. Selections that are traditionally made by mouse clicks are made by visually fixating on a screen icon for a pre-specified dwell time. Coupled with an on-screen keyboard or pictures representing words and messages, as well as text-to-speech voices, the user can communicate messages by selecting pictures or composing text. Eye gaze controlled computers can play a critical role in communication and participation in daily life and society [19]. For example, one study found that patients with amyotrophic lateral sclerosis who used eye gaze controlled computers to overcome communication challenges were generally less depressed and had a higher quality of life compared to a group who only used a phonetic board to communicate [20]. Introducing eye gaze controlled computers at home and at school might enable the child to communicate, express wishes, get support with school work, and participate in social activities to a greater extent [21,22,23,24]. Research exploring parents’ [25] and teachers’ [26] experience of children who use eye gaze technology at home and at school indicates that advanced technology provides hope for children´s futures. At the same time, the hope parent and teachers expressed in connection to eye gaze technology was sometimes far removed from reality and what could realistically happen with support from technology. Howell and Larsen [9] include in their definition that other-oriented hope is uncertain but possible to reach. This raises questions about the relationship between implementation of advanced technology that creates other-oriented hope, the meanings people attach to their other-oriented hope and whether one can hope too much for a person?

The overall goal of this study is to explore and deepen the understanding of other-oriented hope related to advanced assistive technology for children with profound disabilities. The aim is to explore how other-oriented hope could be understood and described in relation to the implementation and use of eye gaze controlled computers for children with profound disabilities and with complex needs.

## 2. Methodology 

A phenomenological-hermeneutical method was chosen to analyse narrative interviews [27], as inspired and modified by the philosophy of Ricoeur’s interpretation theory [28]. The starting point is phenomenological, in the sense that the focus is on the participants’ lived experience and in discovering its meaning structure. The life world is the ordinary world, often taken for granted, that appears meaningful to consciousness. To have a phenomenological attitude means to find the essential meaning structure in the subjective lifeworld. The approach is also hermeneutical in the sense that the essential meanings are expressed through narratives. These stories need to be interpreted to reveal their essential meanings [27].

### 2.1. Participants and Settings

This study was a secondary analysis of interview data collected from two previous studies of teachers and parents concerning eye gaze control technology for children with severe motor impairments and without speech. One study was aimed at describing and exploring what it means to parents when their non-verbal children with severe physical impairments receive a gaze-based assisted technology to use in daily life [25]. The other study aimed to explore teachers’ experiences of using eye gaze-assistive computers with pupils with severe disabilities [26]. These two studies are part of a Swedish longitudinal project where children, together with their parents and teachers, participated in an eye gaze controlled computers project at home or in school from 2010 to 2015. All participants were interviewed twice, once at the time of the introduction of eye gaze technology and later during a follow-up interview seven to 12 months later. The first interview focused on concerns and expectations and the second interview focused on the experience of using eye gaze controlled computers in everyday life (at school and in home). All interviews were in-depth and open-ended [29] but also involved an open dialogue, seeking the informant’s reflections and experiences of using and supporting the child with an eye-controlled computer. In each case, the interviews were held at a location of the participant’s own choice, and the interviewers and the participants had no previous relationship. For detailed information about participants, settings and the interviews, see Borgestig, et al. [25] and Rytterström, et al. [26].

In both these qualitative interview studies there were several expressions of hope related to the eye gaze technology in view of the children’s future possibilities. This other-oriented hope was not explored in greater depth in the previous two studies, but was based on the original data coming from an open narrative approach. The transcriptions were judged to constitute rich data and to be sufficient for a secondary analysis [30]. The data were handled in line with what Thorne [31] describes as analytic expansion and retrospective interpretation. Parents and teachers in relationships with young children with severe disabilities who use eye gaze controlled computers are a limited group, and this in itself is a reason for secondary analysis to maximise the utility of the data [30]. Qualitative data is often a rich and unexploited source of research material. The research question and the methods are also close to the primary research, but focus on other-oriented hope with a new analytical lens. Thus, data from parents’ and teachers’ interviews were used in the present study to answer new questions about other-oriented hope connected to the eye gaze controlled computer, which were not fully understood in the previous two studies. Six parents’ and five teachers’ interview transcripts which described six children´s use of the technology (children aged 5–15 years) were chosen as they contained rich data, and varied and in-depth accounts of other-oriented hope related to eye gaze technology. The parents whose interviews were included in the secondary analyses had a mean age of 43, and comprised four mothers and two fathers. The school staff whose interviews were included had a mean age of 49, comprising three teachers and two teacher assistants, whose experience ranged from nine to 39 years. Four teachers were male and one teacher was female. The reason for choosing parents and teachers of the children was that they were close to the children and all supported the children in use of eye gaze technology. The transcriptions were chosen based on rich data, and on varied and in-depth accounts of other-oriented hope related to eye gaze technology. For the secondary analysis, two transcripts were studied from the first interviews, with the remaining nine transcripts coming from the follow-up interviews.

The six children in this study had cerebral palsy with severe motor impairment and required full assistance in all school-related and everyday activities. Two children were assessed to have normal cognition, three had unspecified cognitive impairments, and for one the level of cognition was unknown. None of them were capable of speech. Their severe motor impairments involved difficulties in controlling any body movement voluntarily, except for eye movements, and all used eye-pointing and facial expressions as modes of communication. An infrared sensor was, in most cases, the only way the children could use to control a computer. The children were provided with an eye gaze controlled computer, a Tobii Technology C12 or P10 [32], with a software program including grids individually adapted with pictures, photos and/or symbols to support each child’s communication, interaction and activities at school.

### 2.2. Data Analysis

Data analysis was conducted in accordance with a phenomenological-hermeneutic method developed by Lindseth and Norberg [27] which involves a progression through three stages. The dialectic movement starts with a naïve reading of the text to grasp it as a whole. To gain a naïve understanding of the meaning of the parents’ and teachers’ hope related to the eye gaze controlled computer, the interview text was read through several times as open-mindedly as possible. In this step, the researcher starts to “guess” and construe the meaning [33]. There is no valuation of what is important or unimportant; rather a phenomenological attitude is adopted which allows the text to speak [27]. The naïve understanding was formulated as narratives that guided the structural analyses.

The next stage emanated from the narratives that emerged from the naïve reading. This was a methodological interpretation in the form of structural analyses in order to understand the text’s sense [27]. The entire interview text was divided into meaning units, which could be part of a sentence, several sentences or paragraphs. This condensation was compared and abstracted to other condensations to create themes and formulate the main narratives. The main narrative and the themes were discussed between the authors to find the most probable interpretation. This included a movement between the naïve understanding and the structural analyses, in order to validate the narratives with the themes. The themes are not a description of the narrative per se but a description of the meanings, beyond those taken for granted in the narratives [34]. The last stage involved a comprehensive understanding. The analysis of the text goes from its “movement from sense to reference” [28], which is a critical in-depth interpretation based on the naïve understanding, structural analyses and hope related to assistive technology as an imaginary act to arrive at a deeper understanding of the concept. To include hope as an imaginary act as a theoretical perspective was motivated from a deeper understanding that hope related to technology was not possible to reach with empirical data alone. The last stage was applied late in the research process so the narratives that guided the structure analysis would not be hidden by theories but instead that hope as imaginary act would illuminate hidden aspects of hope [34].

In all three stages, all the authors conducted the analysis in cooperation. During the whole process the researchers’ pre-understandings were reflected and bridled. This means that pre-understanding could not be bracketed but was instead recognised and reflected upon in each step of the research process. It was a research process that was an alternative to a more hypothesis-driven and reductionist research [35] The result is validated by multiple-method design based on data from two different studies and methods [25,26], and demonstrated reliability using thick and rich data due to in-depth interviews and that all participants were interviewed twice. Demonstrated reliability was also achieved by that data were verified in each analysis step using analysis tables, interview quotations and that the participants represent the phenomenon of exploring other-oriented hope and advanced technology used as assistive technology for children with profound disabilities [36].

### 2.3. Research Ethics

The study was approved by the Regional Ethical Review Board, Uppsala (Dnr 2010 316). The research was performed in accordance with the Declaration of Helsinki [37]. The respondents were aware that the data from the interviews could be used in the overall project and in several articles included in the project. All participants were informed that they could withdraw from the interviews at any time without giving any explanation. 

## 3. Results

The results present first the description of the naïve reading formulated in the three main narratives. This is followed by a structural analysis of the three narratives and ends with a comprehensive understanding. All names in the results are fictional. 

### 3.1. Description of the Naïve Reading

During the naïve reading, an overall picture emerged of how parents and teachers expressed their ever present hope for the children. Three main narratives emerged from the naïve reading. The first narrative was about how the eye gaze technology was an assistive device that actually works, and how the children had already improved with the help of the eye gaze controlled computer. The children could perform different activities and convey thoughts, feelings and desires that had previously not been possible. The second narrative described how the parents’ and teachers’ hope related to the eye gaze technology were connected to the progress already made by the child and what this might mean in the future. Before the implementation of the eye gaze controlled computer, the hope was focused on how the computer can change the child’s everyday situation. After a year of use, the hope relating to present everyday activities was still alive, but there was also a new and growing hope about what the computer could mean for the children in the future:
If we have come this far now, just imagine what it could mean for the future (Parent 4)

They imagined how the child could develop into a more independent individual and handle things by themselves without having to ask for help or refer to others to interpret the needs of the child. Despite great hope being attached to eye gaze technology, the third narrative revealed a hesitation among parents and teachers regarding how far it was possible for the child to develop. The parents and the teachers saw practical and technological difficulties with the eye gaze technology. By recognising the progress the children had already accomplished and what the computer could mean for the children in the future, the parents and the teachers were motivated to encourage the children to use the eye gaze controlled computer:
When you see him today you get hopeful. But you have to remember to look at the long-term perspective. It takes time (Teacher 5).

The parents and the teachers said that they had to be realistic and see things from a longer-term perspective. The future was hopeful but also unsure.

### 3.2. Structural Analyses

The structural analyses of the three narratives that emerged in the naïve reading were analysed through the condensation of meaning units into themes and main narrative. Each narrative and its accompanying themes were named according to the narrative’s central message and are described below. Table 1 provides a summary. 

#### 3.2.1. First Narrative—The Technology Already Makes a Difference

The structural analysis of the first narrative consists of three themes (Table 2).

Parents and teachers describe how the eye gaze controlled computer makes a change (Theme 1). Before the introduction of an eye gaze controlled computer, there were few and insufficient assistive products that could help these children. The parents and the teachers found that these previous assistive products did not work optimally and have not been adapted to the children’s severe disabilities. This has meant that they are unsure of what the child is trying to express and often had to guess what the child wants to do and say.


*“You know, previously it was very much a case of guessing and trying to interpret what Susan wanted to say. Of course, the more we got to know Susan, the better it went, but sometimes it was almost impossible. She got angry and that was very frustrating.” (Teacher 4)*


The parents and teachers describe how these assistive products create frustration and powerlessness since they cannot mediate what the child wants. However, with the help of the eye gaze controlled computer, the child could do things that were previously impossible.

With the introduction of the eye gaze controlled computer, the parents and the teachers notice that the child has made progress and learnt new things in different areas (Theme 2). 


*“If you look back at the time when he started to use the computer, he can now do things that we thought were impossible to begin with.” (Teacher 5)*


After one year, the children have developed the technique for controlling the computer with their eyes. The children are faster at controlling the mouse pointer, more independent in their use of the computer and able to sit for longer at the computer. The parents and the teachers note that this means that the child learns more at school and at home. They also describe how the computer has become a part of the child’s play, in which the child is increasingly able to perform independently.


*“I don’t have to sit as close to him. If he plays a game, I can do something else for a while. When I sat with him, before the computer was introduced, when I went away I knew not much was happening for Johan. Now I know that he can continue to play games or whatever it is he’s doing at the computer.” (Parent 1)*


The children can choose and start computer games or movies by themselves, and this gives parents and teachers a sense of freedom because they do not always have to be immediately at hand. The progress made with the eye gaze controlled computer also relates to the child’s ability to convey needs, feelings and wishes (Theme 3). This progress goes beyond the more educational (knowledge) progress that has occurred. The eye control computer has meant that it is easier for the child to express views, thoughts, needs and desires. Before the introduction of the eye gaze controlled computer, the parents and teacher often needed to guess what the child was thinking and feeling. Through the computer, the child can, for example, convey who they want to be with during school breaks or choose what food they want to eat for lunch.


*“Now we can ask him for example if he feels pain, if he is happy or sad and things like that. Ehh… without the computer you have to guess… So it is good that he has an aid because now he can say exactly what he wants.” (Parent 5)*


The eye gaze controlled computer has opened up a new world for the child and has provided a new opportunity to get to know the child. Through the eye gaze controlled computer, the child can express feelings and experiences that were previously difficult to express. The computer offers new opportunities to understand the child, an understanding that parents and teachers previously experienced as being almost impossible for the child to convey.

#### 3.2.2. Second Narrative—The Technology Points to a Brighter Future

The structural analysis of the second narrative consists of four themes (Table 3).

The narrative shows how the progress already made in everyday life is related to the future, where eye control technology can unlock the child’s future. These imaginings are in contrast to how the child’s situation appears today. Parents and teachers describe how the technology can mean a bright future with new undreamt of possibilities for the child (Theme 1). They imagine that the child will be able to develop his interests, study at upper secondary school level or university level, and might be able to have a professional job:
“But if he succeeds with this, then he can get a job… He wants to become a sports reporter.” (Parent 6)

The hope for new opportunities in education and career choices is related to the consequences of the computer making the child more independent and thus to a greater extent able to shape their own future. The computer is described as a “door-opener” or as an “enabler” of the future, and as “opening up the world” for the child. 

The narrative also relates to how parents and teachers express a hope for future relationships (Theme 2) with other people. When they describe what the future might look like, the computer could give the child a “new language” that will help the child to communicate and make it easier to integrate and communicate with other people:
“That he gets a language so he can express himself… can communicate with almost, yes with anyone.” (Parent 2)

By developing a “language” through the eye gaze controlled computer, there is hope that, in the future, the child could build relationships, make friends or find a life partner:
“He has a sense of humour, it has become so clear now. And imagine what this could mean in the future. Maybe he… Well maybe there is a girl that he can get along with if he, you know, can show more of who he really is?” (Teacher 5)

Due to their severe disabilities, the children have difficulty expressing their own will. This means that parents and teachers sometimes have to guess what the child means and they make decisions that are not always in line with what the child actually wants. In the narratives, the parents and the teachers express a hope that the computer will lead to the child influencing his own life (Theme 3):
“Yes, my thought is that he will become as independent as possible, that he himself should be able to affect his own life. His choice, what he wants…” (Parent 3)

If the child can use the computer to communicate his own wishes, there is a possibility that the child himself can determine his own future and not be dependent on other people.

The parents and teachers have great confidence in the eye gaze controlled computer. Although there are limitations and shortcomings in the technology, there is a belief that technology will improve to give even more possibilities to children in the future (Theme 4):
“Certainly there are things about the computer that could be better. But I think, this is just the beginning. If Alice becomes even better at eye gaze control and if computers develop, well then, it is only your imagination that limits what can happen.” (Teacher 1)

The computer the children use today is described as the first generation of eye gaze controlled computers. The parents and the teachers express their anticipation of the way in which the technology will be improved and become more integrated into the child’s everyday life and a natural part of the child’s future.

#### 3.2.3. The Third Narrative—Living with an Uncertain Future

The structural analysis of the third narrative consists of three themes (Table 4).

This narrative describes how parents and teachers, despite the hope they have for the future, have doubts about whether this will be realised, and this is something they have to live with. The parents and the teachers describe how their hope is surrounded by great uncertainty about what it is possible to achieve (Theme 1). Despite the fact that parents and teachers put their confidence in the eye gaze technology, they also describe the technology as having limitations:
“He (the child) can express things that were not possible before. But the question is how far is it possible to go? Perhaps this is as far as you can go with eye gaze control?” (Parent 4)

The uncertainty is not just related to the technology itself, but also to the child’s own ability to use the eye gaze controlled computer. There is uncertainty about whether the child has the cognitive abilities, strength, endurance or the physical conditions to continue developing the eye gaze control:
“It feels like everything is possible but you don’t know if she is going to manage it. Maybe it doesn’t matter how good the eye gaze technology is. Maybe she can never learn this? Nobody knows. Time will tell what is possible.” (Teacher 2)

The parents and the teachers state that the children’s disabilities are so extensive that an eye gaze controlled computer cannot solve all the problems that the child has and will face. This could involve not only practical problems regarding difficulties in everyday life such as limited mobility and complications but also more general societal aspects:
“This feeling of being left outside is a huge thing… Neither Michael (the dad) nor I dare to hope that the eye gaze computer could overcome those problems.” (Parent 6)

The parents and the teachers also describe not having overly high expectations of the children (Theme 2). Even if there is progress, they are aware that there are and will be obstacles:
“Not believing that everything will go smoothly.” (Parent 3)

Both the parents and the teachers mention that they sometimes want the child to use the computer more than the child does.

The parents and the teachers do not expect this bright future related to the eye gaze controlled computer to come easily or without hard work. The narrative about a bright future is mixed with narratives about whether the hard work will pay off in the future. Parents and teachers describe how the knowledge that the child develops and communicates via the computer develops all the time, but that they have to see everything from a longer-term perspective (Theme 3). They have to remind themselves to take things slowly and view the children’s computer use as a lifelong project. Mastering the eye gaze controlled computer takes time and must be given time:
“It’s a start and not everything has been easy. But you have to start somewhere. You can’t give up.” (Teacher 3)

The parents and the teachers find that to control a computer by eye gaze requires work over time, and no one can say for sure what the result will be or how much the child will be able to achieve. 

### 3.3. Comprehensive Understanding

The three narratives that emerged from the naïve understanding, and that guided the structural analyses, were interpreted based on a framework of hope as imagination within the comprehensive understanding. The parents’ and teachers’ other-oriented hope could be described as going back and forth on a continuum (Figure 1).

At one end of the continuum is the past and the present. As described in the first narrative, “The technology already makes a difference”; both parents and teachers experience the progress the child has already made with the computer at school and at home. This is partly progress with the technology itself as the child becomes more skilled at controlling the computer by eye movements. It is also partly progress with school work or an ability to express feelings and desires. At school, for example, the child could say who they want to play with at recess or describe in detail what kind of food they want to have at lunch. These types of progress are in glaring contrast with the child’s history of struggling to be understood and participate in activities.

At the other end of the continuum the hope is conveyed in technology’s promise about how the child can overcome any obstacles that exist today, as described in the second narrative “The technology points to a brighter future”. The progress that the child has already achieved with the eye gaze controlled computer means that parents and teachers dare to hope and lift their eyes beyond the given situation, imagining what the future holds. The contrast between the past and the progress the child has already accomplished fuels an imagination of a future where the child could be more independent, participating in higher education or employment. This narrative is also related to tremendous confidence in technology, which has unprecedented opportunities to develop in the future. 

This is not a denial of reality or the struggle the children, parents and teachers go through every day. In the third narrative, “Living with an uncertain future”, an uncertainty emerges about whether the imaginary future is possible to achieve, related to both the technology and the child’s physical and cognitive ability. The uncertainty is reflected by parents and teachers when they take responsibility for their doubts about the imaginary future by not having too high expectations or not moving too quickly towards more advanced tasks. 

These three narratives involve a movement from past to present to future. The parents and the teachers connect their struggle and the child’s history with the child’s present situation, and by imagining what the future holds. The parents and teachers do not necessarily believe that the child will be able to do everything in the future, but they imagine a much brighter future. The eye gaze controlled computer makes this promise plausible, although it might otherwise seem impossible. These imaginations break with the present’s limitations and the status quo and point towards other alternatives. This opens up opportunities and a motivation for new ways to understand the present and what is happening in the present. As long as there is a movement along the continuum, there is a self-calibration of the hope that embraces and acknowledges both the present and the imagined future. This interpretation is based on the interviews when they connect the future and the present, at the same time emphasising the parents and the teachers see everything from a longer-term perspective. The technology could therefore be described as a promise of a brighter future, a future that is unfolding in the present and upheld in everyday activities. 

## 4. Discussion

The results show that the eye gaze controlled computer offers a hope for the parents and teachers, and fuels the imagination of a brighter future for the child. That hope is important as previous research has described how hope motivates the utilisation of technology [38]. Earlier research has also found that other-oriented hope may lead to greater life satisfaction, less parenting stress and less depression in parents [5] This study advances knowledge about other-oriented hope and technology in that parents and teachers live with two perspectives: the present perspective, with its obstacles but also progress with the assistive technology, and the imaginary future perspective, with its undreamt of possibilities of what the assistive technology may bring about in the future. There is a movement between these perspectives that seems to function as self-calibration, a movement that has implications for parents’ and teachers’ motivation for using technology. What seems to be unrealistic hope does not become problematic until the movement between the present and the imaginary future ceases. Ceasing this movement can mean that parents and teachers are uncritically creating a vision of the future that does not originate in an understanding of the difficulties that have been and will be experienced. That notwithstanding, there is probably a much greater risk that the movement along the continuum will stop because parents and teachers feel frustration and powerlessness, leading to the difficulties in the present shaping the conditions for the future. For that reason, clinicians should not dismiss the imaginary act, the hope, based on the fact that it is not realistic. Dismissing the imaginary act could undermine the strength and power parents and teachers need in the present for actions. Therefore, this research points to a limitation in defining other-related hope as uncertain but possible [9]. This study shows that parents’ and teachers’ other-oriented hope for their children cannot always be described as realistic or possible to achieve considering the children’s severe disabilities. Nevertheless, as long as their hope for the child is a movement between the present situation and undreamt of possibilities in the future, this research indicates that the movement functions as self-calibration of this hope. The technology creates a promise that may never be realised because of the children’s severe cognitive and physical impairments. However, the hope not only concerns the future, it already exists in the present and helps parents to cope with difficulties, and opens up new alternatives and possibilities to overcome the difficulties the child is encountering today. 

The conviction that this could be is not an escape from reality or a delusion. For these parents and teachers, the imagination of a future is upheld in the present’s everyday activities. Even when there are barriers and obstacles, the imagined future plays a creative and positive role in the present. In the movement between the undreamt of possibilities for the child and the limitations of the present, there is an intrinsic calibration that is continually renegotiated within the person. This means that the other-oriented hope that is mediated in the imaginary act is not static but is in a state of constant change. If a person aims for what they hope for, health professionals need to treat the parents as people who are in a movement between the child’s present situation and an imagination of what the future holds, which means what they express in one moment is not the whole truth. Health professionals need to capture the imagination expressed in the other-related hope and sometimes restore this imagination rather than jumping to conclusions about what is realistic or unrealistic. The model shown in Figure 1 describes other-oriented hope as a continuum between the present and the imaginary future and shows how this creates motivation and action in the present. This model is based on technology introduced for children with severe disabilities, but might also be applicable when technology is introduced for other persons and their next of kin, such as the elderly. This needs to be explored in future research. This model can be helpful for improving health professionals’ understanding of important aspects when introducing advanced technology for patients and their next of kin.

For health professionals, hope is described as a struggle between telling the truth and not destroying hope [39]. This study demonstrates that the other-oriented hope can be described as a negotiation between the parent’s and the teacher’s imagination of a new, undreamt of future for the children, and whether technology can actually realise that imagination. The hope is tied to the human lifeworld with a web of meanings that extends to other values [40]. Behind the imagination of undreamt of possibilities for the children, there might be an extended other-oriented hope that the children will be understood, taken seriously, find their place in the world and be seen. The technology is a catalyst for this hope and carries a promise of a better world for the children. This is important knowledge when new technology is implemented in a health care context. If technology carries a promise of a better world for the child, then those involved need to have the opportunities to express how they view the future and how the technology could enable this hope. It is not always possible to foresee how the implementation of new technology will succeed, but if hope counteracts fear and despair [13], it is important to mediate the creation and maintenance of hope. 

In interactions between health professionals and patients, Mattingly [12] describes an underlying tug-of-war about which story is valid. There is a risk that clinicians will interpret the imaginary acts not primarily as constituting a hope that motivates continued use of eye gaze controlled computer, but rather as a denial and an escape from reality. For Mattingly [12], the imagination is an expression that parents and teachers experience as a moral obligation to establish a dignified life for the child, despite the great difficulties the child comes up against. Therefore, in the implementation phase, clinicians should—in addition to introducing technology, education and technical support—also pay attention to the child’s life situation [15], for example by leaving some leeway for individuals to talk about how they imagine the future, to describe their outlook on life, their priorities, their everyday lives and their routines, and the hope they put in the technology. The meaning they ascribe to the technology cannot be taken for granted. Instead, the implementation of new technology must be adapted and actively developed based on the family’s situation [41]. Also, the design of the technology should be based on a contextual perspective to ensure it is easy to integrate into everyday life, easy to handle and, in addition, mobile, so that it can be used in many different contexts.

The results show how, even after a short period of use, the children have improved in their ability to control the computer, and are capable of things that were previously impossible. These progressions make it easier for parents and teachers to project a positive and improved future, which in turn justifies the continued use of eye gaze controlled computers. A likely conclusion that can be drawn from the study to strengthen the children’s quality of life is the importance of quickly seeing positive improvements in the implementation of new and advanced assistive technology. Imagination also involves acting and functioning in the present. Imagination points to a sense of possibilities each time the parents see improvements in the child’s daily use of the computer. It also suggests that introducing new technology must be preceded by educational efforts to learn to use a gaze based controlled computer, and that prompt technical support for the introduction of technology should definitely be available. 

### Limitations and Recommendations for Further Research

This study follows Lindseth and Norberg [27] phenomenological-hermeneutical approach but, following examples from Salmela, et al. [42], the naïve understanding emphasises the narratives that guide the structural analyses. The parents’ and the teachers’ experiences of the child and the eye gaze controlled computer are embedded in their practical actions, and this experience in daily life involves narrativity [43]. 

The 11 transcripts included in the secondary analysis were chosen from a total of 37 interviews, and were selected because of their rich descriptions of other-oriented hope. Their data was meaning rich and led to a new theoretical understanding of other-oriented hope, which is one approach for if data have reached saturation [44]. That said, a secondary analysis is based on pre-existing interviews and they were not driven by a research question around hope. This is a limitation and it is possible that interviews with a specific aim of elucidating other-oriented hope could reveal new aspects of other-oriented hope and technology. 

Other limitations are that we did not follow the parents and teachers for more than a year. The implementation of the eye gaze controlled computers was preceded by specific training measures and follow-up meetings with the project team. The results of the study should therefore be interpreted on the basis that all participants received support and attention throughout the project. However, as a suggestion for future research, these training measures and follow-up meetings would also be a possible arrangement for implementing advanced assistive technology in a clinical setting. In the analysis of the interviews, the experience of other-oriented hope has been at the forefront, where the results were analysed without making any distinction between the teachers and the parents. Therefore, it is not possible to distinguish between the two groups. The fact that both groups were included in the analysis is motivated by the fact that the two groups can give more variation and richer meanings [35] than only one group giving their experience of other-oriented hope. Future research should interview families and teachers with the intention of exploring other-oriented hope related to both eye gaze technology but also other assistive technology.

Another limitation is that member checking with participants was not possible due to the time that elapsed between the interviews and the secondary analysis. However, the value of member checking is questioned in qualitative research. Morse [36] also emphasises that in establishing validity in qualitative research, the researcher bias can both warp and validate findings. The first author (P.R.) had no experience of professionally working with children with severe disabilities. The other authors (H.H. and M.B.) had long experience of research and professionally working with these children, parents, teachers and also with the eye gaze technology. In the research process we were able to validate and correct each other in an effort to challenge any preconceived ideas but at the same time, the prior knowledge was useful for understanding the context and settings. This approach could be named as ‘bridled openness’ [34] where the research process was to ‘not make definite what is indefinite’, to not draw conclusions that could not be fully supported. The study’s results are also limited to parents and teachers, and further research should include the perspectives of children and clinicians.

## 5. Conclusions

The introduction of new advanced assistive technology creates hope and could be seen as offering the promise of a brighter future, a future that is unfolding in the present and gives motivation in everyday activities. The movement between the child’s present limitations and the imagined future with the support of assistive technology indicates there is an intrinsic calibration that is continually renegotiated within the person. Both the present situation and the hope for the future influence each other and both affect the motivation for using the technology. It is therefore important that clinicians listen to how they imagine the future and understand how hope could strengthen and empower persons in the present to take action and to overcome difficulties. The fact that the imagination of a future is nurtured by progress in present activities stresses the importance of quickly seeing positive improvements in the implementation of new and advanced assistive technology. To see these improvements in present activities, it is possible that the eye gaze controlled computers need to be more mobile and more easily adaptable, so they can accommodate progress and ensure successful implementation, and thereby strengthen a person´s quality of life. It is also suggested that alongside the implementation of eye gaze technology, thorough communication partner training and ongoing support are required.

## Figures and Tables

**Figure 1 ijerph-16-01667-f001:**
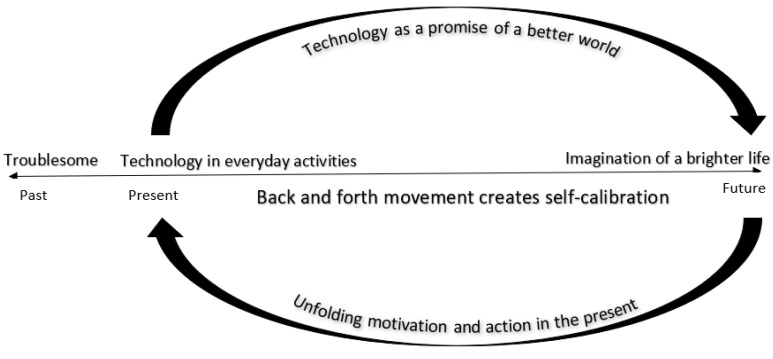
Continuum of the other-oriented hope related to technology.

**Table 1 ijerph-16-01667-t001:** Narratives and themes.

Narratives	Themes
The first narrative: The technology already makes a difference	Theme 1: First time an assistive product makes a change Theme 2: The child has made progress and has learnt new things Theme 3: The child can convey needs, feelings and wishes
The second narrative: The technology points to a brighter future	Theme 1: Hope for new, undreamt of possibilities Theme 2: Hope for future relationships Theme 3: Hope that the child may influence his own life Theme 4: Technology will improve to give even more possibilities to children in the future
The third narrative: Living with an uncertain future	Theme 1: Uncertainty about what is possible to achieve Theme 2: Not having too high expectations Theme 3: Seeing everything from a longer-term perspective

**Table 2 ijerph-16-01667-t002:** Themes and examples of connected meaning units and condensed meaning in the first narrative.

The first Narrative: The Technology already Makes a Difference		
Meaning Units	Condensed Meanings	Themes
*“… tried these head-operated mice and arms and stuff like that. But nothing has been effective… This was like the last chance… now we see something new” (Parent 3)*	Nothing has previously been effective until now	Theme 1: The first time an assistive product makes a difference
*“But with this eye gaze computer, it was the first time that I really believed this could make a difference. And we are already seeing great progress.” (Teacher 5)*	The eye gaze controlled computer makes a difference	
*“For his part, his vocabulary is growing, with the help of the computer. (Parent 2)*	The child can do more	Theme 2: The child has made progress and learnt new things
*“If you look back at the beginning when he started to use the computer, he can now do things that you never thought would be possible to achieve.” (Teacher 5)*	Do things that were not possible before	
*“Now we can ask him for example if he feels pain, if he is happy or sad and things like that. Ehh… without the computer you have to guess… So it is good that he has an aid because now he can say exactly what he wants.” (Parent 5)*	Unlike before, with the computer he can express what he wants	Theme 3: The child can convey needs, feelings and wishes

**Table 3 ijerph-16-01667-t003:** Themes and examples of connected meaning units and condensed meaning in the second narrative.

The Second Narrative: The Technology Points to a Brighter Future		
Meaning Units	Condensed Meanings	Themes
*“We hope that this creates new opportunities before school starts.” (Parent 6)*	New opportunities before school starts	Theme 1: Hope for new undreamt of possibilities
*“I hope he will be able to learn to read and write, for then a whole new world would be opened up.” (Parent 2)*	To open up a new world	
*“Who knows, with the help of the computer he could study at a university. That was absolutely unthinkable before, but now, maybe?” (Teacher 3)*	Unthinkable before but now maybe study at a university	
*“Maybe he could get a job. Just like everyone else. A normal life. You know, working weekdays and then have time off at weekends.” (Parent 1)*	To get a job and a normal life	
*“That he develops a language so he can express himself… can communicate with almost, yes with anyone.” (Parent 2)*	The child can develop a new language and communicate	Theme 2: Hope for future relationships
*“But hopefully it will be so. That he can hold a regular conversation with a person as well.” (Teacher 1)*	In the future the child can have a normal conversation	
*“But who knows? In ten years, maybe he can have a life with many friends and so on. If so, you must be able to speak or communicate in some way. Yes, maybe even move in with someone.” (Teacher 4)*	The child can have friends and someone to live with	
*“(Previously), if it was not too bad, he was pleased with it. I think he can have… yes, if this works really well, it can be more how he wants it to be.” (Parent 2)*	That it will be how the child wants it to be	Theme 3: Hope that the child may influence his own life
*“But that he can decide. He can decide by himself He can say: “now I will tell you this…” Getting to know the freedom.” (Parent 3)*	The freedom to decide for yourself	
*“The technology will, well, it will be better, better and cheaper.” (Parent 4)*	The technology will be better and cheaper	Theme 4: Technology will improve to give even more possibilities to children in the future
*“Even though we have not gained optimal use of the computer, it will develop, because it is almost unlimited.” (Teacher 6)*	The development of technology has no limitations	

**Table 4 ijerph-16-01667-t004:** Themes and examples of connected meaning units and condensed meaning in the third narrative.

The Third Narrative: Living with an Uncertain Future
Meaning Units	Condensed Meanings	Themes
*“He can express things that were not possible before. But the question is how far it is possible to go? Perhaps this is as far as you can go with eye gaze control?” (Teacher 2)*	Perhaps this is how far you can go with the eye gaze computer	Theme 1: Uncertainty about what it is possible to achieve
*“It feels like everything is possible but you don’t know if she is going to manage it. Maybe it doesn’t matter how good the eye gaze technology is. Maybe she can never learn this? Nobody knows. Time will tell what is possible.” (Parent 4)*	Uncertainties about the child’s own ability	
*“This feeling of being left outside is a huge thing… Neither Michael (the dad) nor I dare to hope that the eye gaze computer could overcome those problems.” (Parent 5)*	The eye gaze controlled computer does not solve all problems	Theme 2: Not having expectations that are too high
*“It’s the frustration that it is not always how you… as it was supposed to be.” (Parent 5))*	Not as good as it was initially thought to be	
*“Yes, it’s easy to want too much and too fast.” (Teacher 4)*	Wanting too much	
*“It’s a start and not everything has been easy. But you have to start somewhere. You can’t give up.” (Teacher 3)*	A hard beginning but a start	Theme 3: Seeing everything from a longer-term perspective
*“Not believing that everything will go smoothly.” (Parent 3)*	Takes time

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
