# Peer review of "Hope and Technology: Other-Oriented Hope Related to Eye Gaze Technology for Children with Severe Disabilities"

_ijerph, 2019, doi:10.3390/ijerph16101667_

Round 1

Reviewer 1 Report

This is a really interesting topic and paper which addresses an important and inspiring aspect of the clinical implementation of eye-gaze control technology for children with CP with profound disability.

Overall, it is a paper which is important to publish. It contains a handful of limitations which if declared and justified would enable the paper to be more publishable than currently where they are simply absent. These are covered in the following comments. Addressing the following issues/suggestions would also enhance this useful, relevant and timely paper.

I would suggest that the terms eye-gaze control or variations are used in the title and abstract to assist indexing and to be more reflective of the content of the article. I am unclear why the umbrella term advanced assistive technology is preferred?

The introduction is clear, informative, interesting and provides a useful background to the topic.

Line 37: “This study wants to deepen the understanding of other-oriented hope and advanced technology as assistive technology for children with profound disabilities.” This is usually the place where the aim of the study is articulated clearly and would need to be more specific. A clear aim needs to be added.

Methods. It is not commonplace to do a secondary analysis of qualitative data but the rationale of employing interpretation theory is really interesting, but incomplete. The justification, summarised towards the end of the paper, should go in the methods section eg adapting lines 38-40 on page 11. Our analysis of qualitative data is overlayed with our biases and of course the research questions/aims. Neither of these are articulated.

A lot of our understanding of the background to this study requires the reader to locate the original two references, which is a shame. A little more context would be useful. In particular for this methodology would be the research aims of the original interviews and the relationship of the interviewer to the participants.

There is some description of the characteristics of the children in the original work, but not of the actual adult participants in this study.

The data analysis section is a good description. The methods section needs an explanation/description of the lens through which the data are analysed. How did you allow additional/more detailed/richer themes around hope to emerge. It needs information on who was involved in what processes. There is no discussion of the processes used to justify the credibility of the findings, for instance, did a single researcher complete the analyses? This and issues around triangulation and other methods used ( or not used) to enhance methodological rigour could be mentioned here or acknowledged and justified later in a limitations section.

The paper overall does not address an examination of the researchers role, biases brought to the research and influences on selection of data and analyses – perhaps these could be acknowledged in the limitations section and justified. I imagine that given the time since the interviews, member checking with participants was not completed.

Results section is informative and interesting and well connected to the methods. It is written however in present tense. Results are written in past tense.

The discussion is an eloquently written piece of work which explores the results in depth and addresses clinical implications in a useful and enlightening manner.

Section called “Method discussion”. This section is a weakness in the paper. The first paragraph in this section seems out of place, I can’t see what it adds. In addition, it introduces terminology not previously mentioned. This paragraphs needs reconsideration. The 2nd paragraph in this section should really go as part of the methods as it helps to justify the secondary analysis at an early stage where I, as a reader, were asking these questions. A more descriptive term than “limited group” is necessary – especially as the children were not the study participants in this study. The 3rd paragraph – the first point about follow up for more than a year could be framed as suggestions for future research to add to your understanding of the model and the proposal of self calibration. Another suggestion for future research could be further study which specifically aims to interview families and teachers with the intent or including the intent to explore other oriented hope. The second point about the support and training is really a discussion point – an important one to understand the circumstances under which the participants were interviewed.

There needs inclusion of a Limitations section. It seems important in a limitations section to mention that the interviews were not driven by a research question nor interview schedule which were around hope and may have not drawn out all issues.  This pertains to issues of data saturation which could also be mentioned as a limitation – although you were limited by a pre-existing set of data, what was the experience of data saturation.

The final sentence about mobile and easily adaptable technology – I agree completely with. However, there is not information in the article which supports this. It may be more useful to be clear that you say something like (and this is not well written but to cover the points as an example): To achieve positive improvements quickly and the imaginings of hope with a view to increasing quality of life, early introduction of eye-gaze control technology which is mobile and easily adaptable to accommodate progress, alongside thorough communication partner training and ongoing support are suggested.

Paragraph 1: “other-oriented hope is related to experiencing emotional distress” – please be clear about the direction of the relationship

Page 2/14 line 32. I think it would be useful to clarify that the control group used low tech AAC.  Line 35 – does this research relate to the perceptions of parents and teachers

Line 43: “This is the ordinary world, often taken for granted, that appears meaningful to consciousness.”  It is not clear what “This” refers to and the sentence does not make sense and needs amendment.

There is some inconsistency in the use of the terms “story”, “narrative”, “themes” and “sub-themes” and later pre-narrative” throughout. Narratives generally refer to the content of interviews, what is transcribed. From there themes and subthemes emerge from analysis.

Line 49: “this study is based on…” this is not clear enough. Here is where you need to say that the study was a secondary analysis of data collected from interviews conducted as part of two previous studies…..  If this is not the case, then the reporting is not clear as this is the message I distilled from reading the article.

Page 3/14: line 7: “An auxillary finding..” This needs to be clearer. What is an auxillary finding? Was it a theme which emerged? Was it researcher perspectives? Why was it not identified as part of the qualitative analysis. It is important to be transparent about this. The 2nd sentence in this paragraph does not make sense.

Line 12. Consider: “Thus, data from parents’ and teachers’ narratives were used in the present study to answer new questions about other-oriented hope connected to the eye gaze controlled computer, but which were not fully analyzed in the previous two studies.  Six parents’ and five teachers’ interview transcripts which described six children´s use of the technology (aged 5-15 years) were chosen as they contained rich data, and varied and in-depth accounts of other-oriented hope related to eye gaze technology.”

Line 37: “text TO speak”

Page 4 lines 36: this sentence was not easy to follow – for instance what is continuous text. Suggest: Each narrative and its accompanying themes were named according to the narrative’s central message and described below. Table 1 provides a summary.

Page 7 line1. It is not clear what the subject/object of “This” is.

Page 9: line 20: The following terminology is different than that reported elsewhere in the paper: “To bear the uncertain future”.

Page 11 lines 5 and 8 – Mattingley is incorrectly referenced in both lines.

Author Response

Please, see attached document

Reviewer 2 Report

Review: Hope and technology

Many thanks for the opportunity to review this interesting paper, exploring the hope associated with the implementation of eye gaze technology for children with severe disabilities I believe that this manuscript can make an important contribution to the field. A deeper understanding of the concept of hope and how it relates to action in the present is both of theoretical and clinical interest.

Additional information about the extent of the data the authors base their interpretations on is needed, in order to judge the robustness of their conclusions Also, here and there the authors do not clearly link interpretations to data, creating an impression that they are overextending their findings. Overall, the manuscript would benefit from professional language editing – besides some grammar errors, phrasing is often awkward, resulting in a lack of clarity that detracts unnecessarily from the quality of the paper.

Introduction

The introduction would benefit from a clearer focus on other-oriented hope. Statements as to the lack of conceptual clarity of hope, different types and levels of hope and self-oriented hope diffuse the focus and dilute the authors’ line of argumentation. Similarly, it is not clear how the three perspectives outlined by Olsman (Lines 4-13, Page 2) relate to the study. The perspectives seem to focus on the role of healthcare professionals in responding to the hope expressed by parents. The data of this manuscript does not focus on this aspect, and it is therefore unclear how it relates to the study (see similar challenges in the discussion section). I am also not sure I understand the link between hope and imagination (Lines 8-13). A clearer explanation would be helpful. The third paragraph of the introduction (Lines 14 onwards, Page 2) makes some interesting statements, but, once again, the link of aspects like ‘evidence’ and ‘successful implementation’ to the concept of ‘hope’ is not clear.

The last paragraph gives some background on eye gaze controlled computers. Additional details may be helpful to readers to understand (1) the functioning of the technology (a brief description) and (2) how it facilitates communication and participation - for example, how it can support school work and participation in social activities.  The sentence commencing with ‘The experience of both research…’ is a little unclear. Is the intended meaning that research with teachers and parents has shown that the use of eye gaze technology by children with disabilities provides hope for the child’s future?

Lastly, the aim statement is a little broad and does not clearly show what the study is adding, and why there is a need for deepening our understanding of the issue at hand.

Method

The authors have done a good job overall of explaining the method of analysis used. Consider changing the heading of this section to ‘methodology’. A subheading ‘design’ could be considered for the first section following the main heading. The explanation that phenomenology entails finding the essential meaning structure in the subjective life world’ may need additional explanation.

In my opinion the participants need to be described in slightly more detail – readers will not always have access to the other manuscripts and therefore the current manuscript needs to be able to stand on its own. Please consider adding a table with descriptive detail for each child (e.g., age, educational setting, and details for any assessments – for example, GMFS levels, speech intelligibility assessment). It would also help if authors briefly explained the introduction of the eye gaze controlled computers and the support/intervention given. The interview protocol and process should also briefly be described – what questions were asked? Did the questions differ between the first and second interview? It would also be helpful to know what proportion of interviews had references to other-oriented hope, and how the authors ensured that there was no bias in the selection of interviews.

Authors describe that they included transcripts from both the first and second round of interviews. This means that they received data from two different time points. This is not taken up in the analysis – for example, in Figure 1. In this figure. The ‘present’ may therefore refer to the time point at which the technology was introduced or the time point at which it had been used for a year. This distinction seems important, as the other-oriented hope is portrayed as influenced by experience with the technology. However, much of the results and discussion seem to be written with the second time point as the reference, and no mention is made as to how the other-oriented hope changed from first to second interview. This seems a little problematic.

Data analysis

I was able to get a good understanding of the overall method of analysis used. However, some clarification is needed as to the following:

Who divided the text into meaning units? How many units were found, and how many related to each theme? This information will enable to reader to get a better sense of the extent of the data, and to what extent the interpretations of the authors are supported by data.

The authors make mention that themes and sub-themes were identified – however, the results are not clear on the distinction. The tables only refer to themes, while the text (e.g. Page 5, Line 2) refers to sub-themes). The distinction between themes and sub-themes is therefore not clear.

It is also not clear what is meant by ‘this condensation was compared (to what) and abstracted…”. Who was involved in this step? Was coding done individually and then compared, or as a group, by consensus? This information is important to get a sense of the rigour and trustworthiness of the coding process.

The description of the last step is also not clear. It is unclear what it meant by “the analysis….goes from its movement from sense to reference…” The phrasing is awkward, I would suggest revising to “the analysis progresses from sense to reference’ – however, this still does not clarify what is meant.

Results

Naïve reading: Retain the first sentence of the first paragraph in this section, but integrate the rest of the first paragraph under the three narratives in order to create a better flow and more focus.

Structural analysis: It may be considered to rename the theme ‘technology will improve in the future’ to something like ‘technology will improve to give even more possibilities to children in the future’ so that it is clear that the ‘bright future’ that is expected relates to the child and not to the technology in itself, disconnected from the child.

Tables 2-4 – please give a clearer table heading to each table and use a consistent format. Also clarify whether the table displays themes or subthemes. The first meaning unit reported in Table 2 is not clearly linked to the theme – on face value, it just expresses a hope that the eye gaze technology will make a difference, but it does not specify that it did actually make a difference. Maybe more context would clarify this quote.

The progression from theme to theme in the results section could be clearer. Rather than stating the theme in brackets at the end of an introductory sentence, refer more pertinently to the theme discussed, so that the focus is always clear. For example, start a sentence with ‘The first theme in this narrative dealt with…./focussed on…’

What is the difference between Themes 2 and 3 of the first narrative? Are they not both related to new skills and abilities?

Using the present tense in the results section seems a little unusual, since the parents are not currently being interviewed. I would suggest past tense, but the journal conventions may prescribe differently. Similarly, the use of the present perfect in lines 21-26 (Page 5) seems unusual.

Here and there the phrasing of quotes seems a little awkward. I presume the quotes were translated from Swedish? In that case, the phrasing should be adjusted. For example Line 8, Page 6: ‘What’s this mean…’ should be changed to ‘What does this mean…’ Similarly, it is unclear what is meant by ‘starting point’ (first quote in Table 3). I presume it means something in Swedish but in English it is hard to understand and a clearer translation is required.

The quote in Lines 8-9 (Page 6) is not obviously linked to communication, but rather to agency. Making choices and doing what one wants is not the same as communicating choices and communicating what one wants to do. Maybe more context would help readers to understand why this quote was given as an example of Theme 3.

Line 19, Page 6, and Line 1, Page 7: Starting paragraphs with unclear references such as ‘It shows…’ or ‘This also…’ detracts from the clarity of the manuscript. What does ‘it’ refer to? What does ‘this’ refer to?

The meaning units given in Table 4 as examples of the last theme seem to speak more of perseverance (first example) and of being realistic (second example – possibly related to Theme 2?) – I am not so sure how they relate to the long-term perspective?

Comprehensive understanding: While I really enjoyed reading this section, there were some aspects that seemed hard to link directly to the data. Particularly the labels ‘back and forth movement creates self-calibration’ and ‘unfolding motivation and action in the present’ in Figure 1 (this figure also needs a caption) and the text section describing this aspect (Lines 32-38, Page 9) I found somewhat hard to follow and link to the data. I do think that the references to the past, present and future are seen in the data (although maybe not so clearly captured in the themes), however, the possibility that this creates self-calibration may be more of a suggestion from the authors, rather than something seen directly in the narratives? Does this concept stem from previous literature? Also, how do the data support the statement that hope becomes a source of motivation and behaviour when parents and teachers expose themselves to uncertainty?

Discussion

There are a number of challenges with this section. For one, there seems to be a focus on the health professional and how they should interpret the hope of other team members. This seems to be a clinical application of the results and should not be the main or first focus of the discussion. In my opinion the discussion should first focus on the actual results and how they add to previous literature. (In this regard, the section should be heavily referenced. At present, the first paragraph contains no references.) The discussion should also link up with some of the concepts mentioned in the introduction.

The implications of the study for health care professionals can certainly be discussed as a second point (or even under a separate sub-heading), but the readers should be given a clear indication as to this change of focus in the discussion. Questions such as whether hope is ‘realistic’ are not addressed in this study – authors should therefore rather reference other work on this topic and then discuss the links to the current study. This will counteract the impression that authors are extending their interpretations beyond the actual data.

In Lines 19-20 on page 11, a statement is made that the study shows how children improved in their abilities. This was not the focus of the study and this section needs some revision. Similarly, the last statement in that paragraph (Lines 26-29) is difficult to relate to the actual data.

Methods discussion

I would suggest omitting the first two paragraphs in this section, and then changing the heading to ‘limitations.’ The authors may consider adding any additional limitations such as inclusion of interviews from two time points.

Conclusion

Once again, some statements seem to overextend the interpretation of the data, such as references to the movements between present and future. I would suggest that conclusions stay a little closer to the actual findings of the study. The last statement seems to require a reference.

Editorial suggestions

Please note that these are not comprehensive. As indicated, professional language editing is required to remove some of the awkward phrasing leading to unclear statements.

Page 1

Line 11: Consider replacing ‘strengthen’ with ‘improve’

Line 12: Consider clarifying the phrase ‘illuminates the meanings’

Line 21: The words ‘clinicians’ and ‘healthcare professionals’ seem to be used interchangeably – consider choosing one of these terms and using consistently

Line 27: The clause ‘This study emerges from the hops’ seems unclear. Maybe something like ‘This study seeks to gain a deeper understanding of the hope…’

Line 32: Replace ‘names as’ with ‘term’

Line 36: Specify the comparison made – greater life satisfaction than who? Less parenting stress and less depression than who?

Line 38: Specify the relationship between degree of hope and emotional distress etc. At present it is not clear whether there is a positive or negative correlation.

Page 2

Lines 2-3: Consider bringing in the definition of other-oriented hope earlier.

Lines 14-19: The first few sentences of this paragraph require references.

Page 3

Line 8: Consider replacing ‘developed’ with ‘explored’

Line 9: The second part of the sentence ‘but based on the original data…’ seems unclear and is also lacking a verb.

Lines 20-21: Consider replacing ‘were in total need of assistance’ with ‘required full assistance’

Line 37: Insert ‘to’ before ‘speak’

Page 4

Lines 7-8: Rephrase the sentence to read ‘First, the description of….narratives is presented.’

Table 1: It would help if tables did not run over two pages.

In APA convention, any noun followed by a specific numeral or letter (e.g. Teacher 4, Parent 3, Theme 1) is capitalised. I am not sure if this is a convention also followed for this journal. If so, this should be revised.

Page 6:

Lines 13-14: Consider rephrasing the last part of the sentence (starting with ‘…and understanding that parents….) to make it clearer.

Table 3: Last meaning unit: Was YY a name or a code given to the participant?

Page 7

Lines 18-19: The pronouns ‘his’, ‘himself’ and ‘his’ should be changed to ‘his/her’, ‘him-/herself’ and ‘his/her.’ This should also be revised in other section sof the manuscript. Alternatively, the plural (children) can be used with the accompanying pronouns ‘they, themselves’ etc.

Line 27: Replace the word ‘see how’ with something like “express their anticipation of the way in which…’ Also change ‘become’ to ‘becomes’

Table 4, fourth meaning unit: Maybe add some context as to what ‘it’ refers to.

Page 8

The quote about not believing that everything will go smoothly seems to be provided twice – once in the table and once in Line 23. However, the wording is not identical (possibly due to translation).

Line 22: Change ‘if there are’ to ‘if there is’

Line 24: Change ‘have experienced’ to ‘mention’

Line 25: Omit the word ‘what’

Page 9

Line 30: Replace ‘present’s limitations’ by ‘limitations of the present’

Throughout the manuscript: Replace ‘undreamt possibilities’ with ‘undreamt of possibilities’

Page 10

Line 33: change the order of ‘maintenance and creation’ to ‘creation and maintenance’

Page 11

Line 21: Consider replacing the phrase ‘create their imaginary stories’ to something like ‘project a positive and improved future’ to avoid the impression that the future hope is fictional.

Lines43-44: Replace ‘taken into account’ by ‘interpreted’

Page 12

Line 8: The phrase ‘is in transition with each other’ seems unclear – what is meant here?

Author Response

Please se attached document

Reviewer 3 Report

The paper discusses the problem of eye tracking usage as an assistive technology for disabled people. The discussion is conducted regarding the hope and expectations of obtaining better conditions of life. This analysis is the continuation of the previous research and was performed based on the experience of parents and teachers of some disabled children. 

Unfortunately, I could not find any contribution, which could be considered as new or having an impact on eye-tracking technology development, especially in such applications. They have already been known, as well as their impact on improving people life. 

Thus, I can't see what I and others can learn for the presented analysis.  

Round 2

Reviewer 2 Report

Many thanks for the opportunity to review the revised version of this manuscript. The authors are to be congratulated on their thorough and comprehensive revisions. The manuscript reads clearer now and the aims and discussion are well aligned. The final phase of the analysis is also well-explained. Thank you for the various editorial revisions that add to the overall clarity.

I only have some limited additional comments – some are merely editorial while others relate to content.

Introduction

Page 2, Line 73: Consider replacing ‘and without speech’ by ‘with limited speech’ or ‘with little or no functional speech’ since AAC is not only of value to those who have no speech whatsoever.

Many thanks for adding an explanation of the workings of the eye gaze technology. I want to suggest some edits to this to improve clarity, as I think that the eye gaze is the body movement that replaces hand movements  - so eye gaze does not replace the mouse  because the mouse is the technology – the eye tracker or infrared sensor replaces the mouse and the eye gaze (sensed by the eye tracker) replaces the hand movements (that would be sensed by the mouse or keyboard). I hope that makes sense. I would therefore suggest rewriting the highlighted section on Page 2, lines 74-77as follows:

“An infrared sensor tracks the movement of the eyes and translates this into cursor control. In this way, the user can control the position of the cursor on the computer screen through eye movements, without the aid of a mouse. Selections that are traditionally made by mouse clicks are made by visually fixating on a screen icon for a pre-specified dwell time. Coupled with an on-screen keyboard or pictures representing words and messages, as well as text-to-speech voices, the user can communicate messages by selecting pictures or composing text.”

The terms ‘eye gaze computer’ and ‘eye gaze controlled computer’ are used interchangeably. Consistency in terminology is recommended. I prefer the second term as it seems more descriptive.

At times the plural ‘hopes’ is used (e.g., Page 2, lines 49 and 65). The singular is more commonly used in English and therefore I suggest changing this to the singular noun, ‘hope.’

Thank you for clarifying the comparison in the study of persons with ALS using eye gaze controlled computers. For further clarity, I suggest adding the phrase ‘to overcome communication challenges’ after the phrase ‘eye gaze controlled computers’ (Page 2, Line 80). Also, consider changing the last part of the sentence (‘with only a phonetic board…computers [20].’) to ‘who only used a phonetic board to communicate [20].’

Page 2, Line 87: Change ‘away’ to ‘removed.’

Page 2, Line 88: Delete the word ‘of.’

Page 2, Line 90: Change the word ‘implement’ to ‘implementation of’

Page 2, Line 91: Replace the last part of the sentence (‘… if the hope…a person?’) with ‘whether one can hope too much for a person.’

Page 2, Line 94: To clarify the goal a bit further, I would suggest changing the phrase ‘and advanced technology used as assistive technology’ to ‘related to advanced assistive technology’

Methodology

Many thanks for the inclusion of additional details on participants. This is helpful. I believe this information is sufficient.

Page 3, Line 116: The term ‘gaze-based assisted technology’ seems to be another term for describing eye gaze controlled computers. I would suggest remaining with the same term for the sake of clarity.

Line 138: Remove the word ‘but’

Line 142: Rather than ‘The included parents…’ I would suggest wording it as ‘the parents whose interviews were included….’ Similarly, ‘The included school staff’ (Line 143) should be reworded as ‘the school staff whose interviews were included…’

Line 157: It is a little unclear what you mean by ‘An eye control device’ – is this the whole computer or the infrared sensor? Similarly, in Line 159 the term ‘gaze based assisted technology’ is used again – is this the eye gaze controlled computer? I suggest consistency in terms used.

I appreciate the author’s explanation that the last phase of the analysis does indeed go beyond the empirical data.  I still had a bit of difficulty following the sentence in lines 185 – 187. Based on the authors’ explanation in the cover letter – could this be reworded to something like  “In this last phase, therefore, the authors conceptualised (from a theoretical perspective) hope related to assistive technology as an imaginary act along a time line to arrive at a deeper understanding of the concept.” (If you prefer omitting the time line part I am happy for you to do so.)

Line 192: Consider replacing ‘research’ with ‘process’

Line 193: Replace ‘was not possible to bracketing’ with ‘could not be bracketed’

Lines 195 – 197: The sentence commencing with ‘The result….’ is unclear. Clarify particularly what you mean by ‘multiple-method design,’ how you ensured that data was indeed ‘thick and rich’ and how data saturation was ensured.

Heading of Tables 2-4: Change the word ‘example’ to ‘examples’

Table 3: Remove the horizontal grid line under the 3rd meaning unit.

Line 336: Add ‘with’ after the words ‘not always in line’ (should read ‘not always in line with’)

Line 337: Change ‘influence’ to ‘influencing’

Line 352: Change ‘eye gaze control computers’ to ‘eye gaze controlled computers’ for consistency.

Line 424: Add ‘movements’ at the end of the sentence (after ‘eye’).

Line 425: ‘Learning aspects’ is vague – can you describe more clearly what you mean here?

Thank you for the explanation of the third analysis phase – I am better able to follow it now and see how it links to the data. I think my one confusion still relates to the last sentence, where it is stated that by parents and teachers exposing themselves to uncertainly, hope becomes a source of motivation and behaviour. I think my confusion relates to the dual description of the future – the second narrative visualises a bright future, while the third narrative visualises it as uncertain. I am not sure that it is particularly the ‘uncertain’ part of the future that motivates parents? How do we know it is not the ‘bright’ part? Do quotes like “You have to start somewhere” not suggest that parents and teachers act in spite of uncertainly, not because of it? And that uncertainty (like wanting too much too fast) could moderate present behaviour (the self-calibration referred to)?  It may be beyond the scope of the paper to explore this in greater detail, and in that case the authors may consider omitting the last two sentences of this section (Lines 457 – 460).

Dicussion

Line 473: Replace ‘importance’ with ‘implications’

Line 482: Insert the word ‘in’ before ‘defining’

Line 536: It is not clear to me what ‘their imaginary act’ refers to – can you clarify this? Do you mean the future they imagine for the child? The use of this phrase in Line 605 of the conclusion also needs to be clarified.

Line 552: By ‘education efforts’ – do you mean training in the use of the technology? Maybe this could be clarified.

Line 553: I would suggest omitting ‘to create opportunities,’ since I think technical support is more about enabling and sustaining the use.

Thank you for adding the limitations section. Since recommendations for future research are also given here, you may consider rewording the heading to ‘limitations and recommendations for further research’

Author Response

Attached are our comments. Thank you very much!
